# Fatigue among Greek Parents of Children with Autistic Spectrum Disorder: The Roles of Spirituality and Social Support

**DOI:** 10.3390/healthcare12040455

**Published:** 2024-02-10

**Authors:** Eugenia Halki, Maria Kapiri, Sotirios Plakas, Chrysoula Tsiou, Ourania Govina, Petros Galanis, Victoria Alikari

**Affiliations:** Post Graduate Program “Management of Chronic Diseases- Neurosciences”, Department of Nursing, University of West Attica, 12243 Athens, Greece; euhalki@gmail.com (E.H.); stkts23@gmail.com (M.K.); skplakas@uniwa.gr (S.P.); ctsiou@uniwa.gr (C.T.); ugovina@uniwa.gr (O.G.); pegalan@nurs.uoa.gr (P.G.)

**Keywords:** autism spectrum disorder, fatigue, parents, social support, spirituality

## Abstract

The high demands of caring for and raising a child with autism spectrum disorder on a daily basis may lead parents to physical and mental fatigue. This study aimed to assess the effect of social support and spirituality on the fatigue of parents with children with autistic spectrum disorder. A cross-sectional study with a convenience sample was conducted in Schools of Special Education in Attica (Greece). The sample consisted of 123 parents who completed The Fatigue Assessment Scale (FAS), the Multidimensional Scale of Perceived Social Support (MSPSS), and the Functional Assessment of Chronic Illness Therapy Spiritual Well-Being Scale (FACIT Sp-12) to measure the levels of fatigue, social support, and spirituality, respectively. The Pearson correlation coefficient was used to investigate the relationship between the quantitative variables. To study the effect of social support and spirituality on fatigue, multivariable linear regression was applied. The mean age was 47.3 years old, 81.3% were women, and 38.9% stated “Close/Very close faith toward God”. Higher levels of total MSPSS and FACIT Sp-12 were associated with lower total FAS (r = −0.50, *p* < 0.001 and r = −0.49, *p* < 0.001, respectively). Social support and spirituality were significant predictors of fatigue.

## 1. Introduction

Caring for a child with autism spectrum disorder (ASD) is an extremely stressful experience, leading to a burdened daily life for both the parents and the rest of the family environment. The variety of problems that accompany ASD can cause a dramatic wave of changes in the lives of parents by affecting their emotional, social, and family lives in a multifaceted way. The high demands of raising a child with ASD lead to physical and mental fatigue for parents, as opposed to parents with children of normal development [1,2].

The perception of fatigue is characterized as multidimensional, subjective, and unpleasant, an experience that is difficult to define. This fact is attributed mainly to the subjective nature of the concept, and as a result, several different definitions and measurements arise [3]. One of the predominant emotions experienced by parents of children with ASD is complete exhaustion, which is not just related to the physical fatigue that a parent may feel from daily obligations but also to the inability to regulate autistic behavior [4]. Parents of children with ASD show higher levels of fatigue than parents of children with normal development. In cases where children have sleep disorders, nocturnal awakenings, and intense hyperactivity during the day, parents report severe symptoms of physical exertion, but also loss of control [5]. Fatigue experienced by parents of children with ASD is associated with high levels of stress [6], poor quality of sleep, inadequate social functioning, and poor physical and mental well-being [7]. In addition, maternal fatigue and stress can enhance children’s problem behaviors and, in turn, increase the levels of fatigue that parents already feel [1]. According to the literature, it seems that mothers experience child behavior problems in a different way than fathers [8]. Mothers of children with ASD encountered higher levels of distress due to their child’s behavior, care, anxiety, and inadequate communication abilities. Conversely, fathers are primarily impacted by distressing events unrelated to their child’s disability, such as concerns regarding their professional trajectory or the family’s financial stability [9].

The arrival of a child with ASD affects family relationships both between their own members and the wider social environment [10]. Parents believe that their experiences of raising a child with a developmental disorder are different from those of their relatives and friends who are raising normally developing children. Thus, they are differentiated from other parents and, therefore, feel lonely and isolated [11]. To overcome this problem, the parents’ access to various social support resources is an extremely beneficial coping strategy against the stressful situations helping them to cope more successfully with the demands of their child’s raising [12]. Support is provided by formal (e.g., the state or related organizations that aim to address the needs of the child with ASD and his family) and informal support resources (the wider family and friendly environment of the family [13]. The material help of family members may not be as important as their understanding of ASD and the acceptance of the child with ASD [14]. The perceived social support from various sources such as family or friends seems to be associated with lower levels of stress, and higher levels of quality of life, especially for mothers of children with ASD [15]. Therefore, it seems that social support has been associated with enhancing the mental and physical resilience of parents and consequently with improved and better quality of care for their child [16].

Spirituality is a multidimensional concept, which is difficult to define because of its different aspects. Quite often, the term spirituality is confused with religiosity. Religiosity is related to the acceptance of specific ritual practices and beliefs within an organized religion [17]. In contrast, spirituality is more as it constitutes a human phenomenon that potentially exists in all people and thus differs from religiosity. Essentially, spirituality is about a personal search within and outside of a religious context. This fact also explains that the spiritual life is not a privilege only for people who follow a religious doctrine [18]. The international literature emphasizes the importance of understanding parents’ spiritual beliefs, values, and priorities as the cornerstone of compensating for the negative effects. Spirituality reflects relationships with ourselves, with other people, with God, and with nature [19], while at the same time being associated with a deep sense of peace and satisfaction [20]. Moreover, recent research has concluded that high levels of spirituality in parents of children with special educational needs or wider disabilities are strongly linked to improved dimensions of mental health. Spirituality strengthens them for coping with difficult psychosomatic situations in which they feel pressure under the weight of obligations, fear, anxiety, despair, shame, or even depression [21]. Through their involvement with faith, parents may draw strength to face adversity and adopt a more optimistic approach to life and they may come to experience relief from some of their emotional challenges through prayer [22]. Most parents, through spirituality, can reconsider their perceptions of their child’s disability and turn it from a personal tragedy into a divine gift. They feel that the disability of their children is a blessing from God and that it was given to them by a higher power because of the parental skills they have [23].

As far as it is known, no previous research, not only in Greece but also worldwide, has studied these three variables in parents of children with ASD. Therefore, in order to fill the gap in the literature, the aim of this study was to explore the levels of fatigue, spirituality, and social support among parents of children with ASD. In addition, the relationship between these variables as well as the effect of spirituality, social support, and demographic characteristics on parents’ fatigue was studied. According to the above literature, it was hypothesized that spirituality and social support would be significant predictors of fatigue.

## 2. Materials and Methods

This is a quantitative, descriptive, cross-sectional study conducted between September and November 2020. The cross-sectional nature of this study is supported by the literature among different populations [24,25,26].

The subjects of this study were parents of children with functioning autism who attend three Schools of Special Education (Secondary and High Schools) in Attica (the most populated county of Greece), Greece. The selection of this sample was based on the ease of access and approach of this population (convenience sampling) as the researchers are school nurses. The questionnaires were completed: (i) in the morning after the children were handed over to the teachers, and (ii) at noon while the parents were waiting for the finishing of school. Thus, the children could not see the questionnaires. Each parent was individually given a desk in a separate room where each parent could answer the questions without one parent being influenced by the other. The questionnaires were completed by one or both parents. The inclusion criteria were: (a) be parents of children with functional autism who were attending Schools of Special Education of Attica, (b) be able to read and understand the Greek language, (c) give consent to participate in this study, and (d) be time and space oriented. Parents with cognitive or psychological disorders, vision loss, and parents of children with other physical or mental disorders such as epilepsy, intellectual disability, and attention-deficit hyperactivity disorder were excluded from this study since these disorders are the subject of different studies. All 140 parents (both mothers and fathers) were recruited in this study of which 135 were eligible. Of these, 12 parents did not agree to take part in this study. Finally, 123 questionnaires were completed. The questionnaires were provided by the researchers who are school nurses. Each parent completed the questionnaires once.

The current study complied with the fundamental ethical principles governing the conduct of research. The permission to collect personal data was secured by the Institute of Educational Policy (http://iep.edu.gr/en/ accessed on 7 January 24) (Approval No: 50154/Δ3/30 April 2020). A meeting of the school teachers’ association was held and after receiving positive written suggestions, the questionnaires were provided to the parents and were completed by the parents who wished to participate in this study. Written informed consent was approved by participants. The confidentiality of the information concerning the parents and students of the school was maintained. The parents were informed that the security and anonymity of the relevant material would be preserved and that the results would be used exclusively for the purposes of this research and only by the researchers.

To assess fatigue, social support, and spirituality, the following scales were used:

The Fatigue Assessment Scale (FAS) is structured by 10 items in a five-point Likert scale (1 = never to 5 = always) and examines the levels of perceived fatigue. The scores from the ten items were summed, with the total scores ranging from 10 to 50. For the extraction of the score, the answers of the participants were added. Participants with a score < 22 were classified as “non-fatigued”, 22–34 as “fatigued” and 35–50 as “extremely fatigued”. In this study, the total score was used (ranging from 10 to 50). Five questions are related to physical and five to mental fatigue [27]. It takes two minutes to complete. In addition, studies report the internal consistency and reliability of the FAS both in healthy (Cronbach’s alpha 0.90) [27] and in patients (Cronbach’s alpha 0.88) [28]. It has also been used among parents of children with ASD [1]. The psychometric properties of the Greek version have been tested in chronic disease patients with a Cronbach’s alpha of 0.761 [29] and 0.825 [30].

The Multidimensional Scale of Perceived Social Support (MSPSS) [31] consists of 12 items, which are answered on a seven-point Likert scale (1 strongly disagree to 7 strongly agree). The tool evaluates three sources of social support: Family, Friends, and Significant Others. Each of the above sources is evaluated based on 4 items. The total score ranges between 1 and 7 and is obtained from the sum of the scores and divided by the number of items. The high score reflects higher levels of perceived social support. It is short, easy to use, and understandable even in low-educated populations. In addition, as reported in the research of Zimet et al. [31], the MSPSS shows good internal consistency in different groups of subjects. The scale has been used in Greek patients [32] and healthy populations [33] with very good internal consistency according to Cronbach’s alpha (0.93) [34].

The Functional Assessment of Chronic Illness Therapy-Spiritual Well-Being Scale-12 non-illness (FACIT Sp-12) adapted to the general population is a self-administered questionnaire constructed in 1990 [35] as a short tool for assessing three domains of spirituality: Peace, Meaning of Life, and Faith. It is structured by 12 items on a five-point Likert scale (0 = not at all to 4 = a lot). The questions concern the period of the last 7 days. The total score is derived from the sum of the answers, with the highest scores indicating higher levels of spirituality. The tool has been translated into several languages, including Greek with a Cronbach’s alpha of 0.77 [36].

Finally, data related to sociodemographic characteristics were recorded.

Categorical variables are described using absolute (*n*) and relative (%) frequencies, while quantitative variables are summarized by their mean, standard deviation, minimum, and maximum values. When exploring the association between two quantitative variables that exhibit a normal distribution, the Pearson correlation coefficient was utilized. We considered spirituality, social support, and demographic characteristics as the independent variables. Parents’ fatigue was the dependent variable. First, we conducted bivariate analysis between the independent variables and the dependent variable. Then, variables with a *p*-value < 0.2 in the bivariate analysis were included in a multivariable linear regression model with fatigue scores as the dependent variables. We applied the backward stepwise method to identify statistically significant relationships (*p* < 0.05). In the multivariable models, we present unstandardized and standardized b coefficients, 95% confidence intervals, and *p*-values. Moreover, we performed one-way analysis of variance to investigate the impact of closeness to God with continuous variables. Due to limited numbers, we merged the categories “Not at all close faith toward God” and “Too little faith toward God” and the categories “Close faith toward God” and “Very close faith toward God”. Statistical significance was set at the level of 0.05. Data analysis was applied using IBM SPSS 21.0 (Statistical Package for Social Sciences, SPSS Inc., Chicago, IL, USA).

## 3. Results

The study population included 123 participants. The mean age was 47.3 years, 81.3% were women, 77.2% were married, and 20 (18.6%) participants were married to each other. A percentage of 95.9% were Orthodox Christians and 38.9% stated that they feel “Close faith toward God”/Very close faith toward God”. The sociodemographic characteristics of the parents are presented in Table 1.

Regarding the levels of fatigue, social support, and spirituality, the mean values of the three scales indicated moderate levels of total fatigue (mean 28.41, SD = 7.51), relatively high levels of total spirituality (mean 31.51, SD = 8.73), and very low levels of total social support (mean 5.16, SD = 1.12) (Table 2). Participants appeared to have experienced higher levels of physical fatigue (mean 20.81, SD = 5.80) compared to mental fatigue (mean 8.92, SD = 1.60). The majority evaluated positively the dimensions of spirituality and especially Peace (mean 12.91, SD = 3.01). The most important source of social support was Significant Others (mean 5.23, SD = 1.20).

As far as the correlations between the three variables are concerned, the results showed that both social support and spirituality were negatively associated with fatigue. Table 3 shows that higher levels of total FACIT Sp-12 were associated with lower levels of total fatigue (r = −0.49, *p* < 0.001). Also, higher levels of MSPSS were associated with lower levels of total FAS (r = −0.50, *p* < 0.001). Negative correlations were also observed between all the dimensions of FAS and all the dimensions of MSPSS and FACIT Sp-12. In addition, higher levels of total MSPSS were associated with higher levels of total FACIT Sp-12 (r = 0.51, *p* < 0.001). Also, positive correlations emerged between all the dimensions of MSPSS and all the dimensions of FACIT Sp-12.

After bivariable analyses, statistical relationships emerged at the level of 0.20 (*p* < 0.20) between independent variables (spirituality, social support, and demographics) and total fatigue and its dimensions. For this reason, multivariable linear regressions were applied with total FAS and its dimensions as dependent variables and FACIT Sp-12, MSPSS as independent variables along with demographics. It is observed that Peace and support from Significant Others can positively affect total fatigue since higher levels of Peace were associated with a lower total FAS Score (*p* < 0.001) and higher levels of support from Significant Others were associated with a lower total FAS Score (*p* = 0.008). Also, gender (female) had a significant influence as mothers had a higher total FAS Score than fathers (*p* = 0.001) (Table 4). One-way analysis of variance confirmed that “Close faith toward God” was not related to total FAS (*p* = 0.759). The mean total FAS Score for those with “Not at all/Too little close faith toward God” was 29.21 (SD: 8.33), 28.20 (SD: 7.65) for those with “Little close faith toward God”, and 27.95 (SD: 7.00) for those with “Close/Very close faith toward God”.

In addition, investigating physical fatigue as dependent and FACIT Sp-12, MSPSS as independent variables, a positive effect on fatigue emerged since the higher levels of Peace were associated with lower physical fatigue (*p* < 0.001). Also, mothers experienced higher physical fatigue than fathers (*p* = 0.002) (Table 5). One-way analysis of variance confirmed that “Close faith toward God” was not related to physical fatigue (*p* = 0.885). The mean physical fatigue score for those with “Not at all/Too little close faith toward God” was 21.11 (SD: 6.36), 20.57 (SD: 5.90) for those with “Little close faith toward God”, and 20.79 (SD: 5.49) for those with “Close/Very close faith toward God”.

Similarly, the high level of social support from Family positively affects mental fatigue since an increase in support from Family was associated with an increase in the mental fatigue score (*p* < 0.003). A high educational level was associated with an increase in mental fatigue score (*p* < 0.008), and mothers experienced higher levels of mental fatigue than fathers (*p* = 0.01) (Table 6). One-way analysis of variance confirmed that “Close faith toward God” was not related to mental fatigue (*p* = 0.810). The mean mental fatigue score for those with “Not at all/Too little faith toward God” was 9.01 (SD: 1.72), 8.89 (SD: 1.80) for those with “Little faith toward God”, and 8.81 (SD: 1.50) for those with “Close/Very close faith toward God.

Participants with a lower educational level and a higher “Close faith toward God” had higher scores of total FACIT Sp-12 (β = −1.61, CI: −2.60 to −0.51, *p* = 0.005 and β = 2.50, CI: 1.41 to 3.72, *p* < 0.001, respectively) (Table 7). One-way analysis of variance confirmed that “Close faith toward God” was related to total FACIT Sp-12 (*p* < 0.001). The mean total FACIT-Sp-12 Score for those with “Not at all/Too little close faith to God” was 27.01 (SD: 8.82), 30.90 (SD: 7.26) for those with “Little close faith toward God”, and 35.17 (SD: 8.55) for those with “Close/very close faith toward God”.

Regarding the differences between mothers and fathers, mothers scored higher (mean 29.65, SD: 6.83) on the total FAS than fathers (mean 22.96, SD: 6.84), (*p* = 0.001, Hedge’s g = 0.99), higher on the physical fatigue scale (mean 21.61, SD: 5.55) than fathers (mean 17.00, SD: 5.82), (*p* = 0.001, Hedge’s g = 0.83), and also higher on mental fatigue (mean 9.16, SD: 1.66) than fathers (mean 7.85, SD: 1.60) (*p* = 0.001, Hedge’s g = 0.81).

## 4. Discussion

The current study was conducted among parents of children with ASD and investigated the effect of spirituality and social support on the levels of fatigue. This study is significant, as the lack of spirituality and social support may negatively affect parental fatigue leading to inadequate care for children with ASD [37].

According to the results, the parents experienced moderate levels of total fatigue and spirituality and very low levels of social support. Regarding fatigue, participants reported higher levels of physical fatigue and lower levels of mental fatigue. This finding is in line with the findings of the empirical study of Giallo et al. [1]. The moderate levels of parental fatigue in this study seem to be proportional to the severity of the cognitive and behavioral deficits of ASD considering that the sample of this study consisted of parents of children with functioning autism. In terms of spirituality, parents reported higher spirituality based on the dimensions of Peace, followed by Faith and Meaning of Life. Also, the dimension of Significant Others was the most important source of social support followed by Family and Friends. Nevertheless, other studies have identified Friends and Family as the most important sources of social support [38,39].

As far as the relationships between the three variables are concerned, it emerged that the higher total social support and its dimensions were associated with lower total fatigue and its dimensions. Similar findings were presented by Ardic [40], who observed a significant prevalence of exhaustion in parents receiving mitigated social support. In addition, as shown in other studies [37], the perceived social support received by parents of children with ASD appears to be associated with mental and physical well-being, resilience, and higher quality of life. Higher levels of social support were negatively associated with fatigue, stress, and depression in parents of children with ASD. Thus, these results reflect the importance of informal support networks, as an important strategy for dealing with the mental and physical effects of ASD [37].

Also, a negative correlation was found between fatigue and spirituality as the higher the spirituality levels the lower the fatigue levels. This finding is in line with the finding of a study [41] in which mothers of children with mental disabilities who took spiritual self-care training experienced a reduced burden of care. There is a sense that in the course of time, parents change the way they perceive their child’s disorder and the world. It has been reported that parents of children with disabilities embrace a wide range of positive change, seek answers to questions such as the meaning and purpose of life [42], and acquire personal gifts such as strengthening religious beliefs and greater appreciation even for simpler things in life.

From the findings of this study, it seems that the higher total social support, support from Significant Others, Family, and Friends was associated with higher spirituality based on the dimensions of Meaning of Life, Peace, and Faith. In a previous study [43], the negative correlation between spirituality and social support was attributed to the fact that parents who experienced social isolation were more likely to seek spiritual support. In addition, spiritual pursuits were used more as a means of escape from the daily difficulties and burden of caring for a child with ASD. However, a positive correlation has been observed between social support and spirituality among different populations [44]. This relationship may be attributed to the fact that spiritual beliefs and religious behaviors encourage involvement in social support.

Concerning the effect of spirituality on fatigue levels, it was found that spirituality plays a significant positive role in mental and physical fatigue. Parents with higher levels of Meaning of Life, Peace, and Faith showed lower levels of total, physical and mental fatigue. Similarly, researchers [18] observed higher levels of resilience, sense of coherence, and adjustment in parents of children with ASD who had received some spiritual lessons. Similar results have been presented in previous studies [1,45], emphasizing the significant role of spiritual well-being in the improvement of various dimensions of the psychological sphere such as anxiety, lack of satisfaction, and depression. It seems that parents, through spirituality, face the problem with a positive attitude as they focus more on the positive dimensions and contributions of their child with ASD [46]. This fact may be able to alleviate perceived fatigue.

Regarding the effect of gender on fatigue levels, the present study shows that mothers experience more physical and mental fatigue than fathers. This finding seems to be consistent with other studies [47] which have shown that mothers experience the care of burden and symptoms of fatigue more often than fathers. In addition, according to Nacul et al. [48], fathers seem to be influenced more on a mental level, and women on a physical level. The strategies that parents apply in order to meet the requirements of the parental role differ between mother and father due to particular individual characteristics and differences in family circumstances. The intense fatigue among mothers is probably attributed to the mother’s sense of guilt, the concern for the child’s excessive dependence on the family, and the lack of resilience which leads to adjustment difficulties [49].

Finally, according to other results from the present study, it appears that the higher degree of religiosity and the low level of education had a positive effect on overall spirituality. Regarding religiosity, other studies [20] have suggested that engaging in metaphysical transcendence, spiritual beliefs, and prayer alleviates emotions, improves, and promotes the mental health of parents of children with ASD or other special educational needs.

The sample of this study (convenience sampling) came from schools in Attica; therefore, the generalization of the results may be subject to relevant restrictions. In addition, although the sample size was partially satisfactory, mothers appeared to be the vast majority of participants, which raises significant concerns about the representativeness of the sample. Also, the period of completing the questionnaires (COVID-19 pandemic) may have influenced the objectivity of the views expressed by the parents. However, we should note that the research process was not greatly affected by the COVID-19 pandemic as the special schools in Greece at that period operated in person and not online. Also, the cross-sectional nature of this study and the inability to establish a cause-and-effect relationship could be a limitation. In addition, another limitation of this study is that the researchers and participants knew each other, and therefore social desirability may have played an important role [50]. Future research is suggested that will include clinical data on children and their association with parental fatigue.

## 5. Conclusions

In conclusion, the present study recorded moderate levels of fatigue and spirituality but also low levels of social support for parents raising a child with ASD. Subsequently, significant negative correlations were observed between fatigue and social support, fatigue and spirituality, and a positive correlation between social support and spirituality. The effect of gender seems to be particularly important as mothers of children with ASD seem to experience fatigue to a greater extent than fathers. The results suggest that social support and spirituality help to reduce perceived fatigue and therefore strengthen the resilience of parents of children with ASD. However, the emphasis on personalized spiritual care remains an important dimension. School nurses can also encourage parents to express their spiritual beliefs and needs as well as their inclusion in organized groups or support and mutual aid networks between parents. The process of acquiring adjustment mechanisms that parents can use to manage the exhaustion caused by their child’s stressful demands may be very useful.

Social policies that focus on psycho-educating parents about fatigue and its potential impact on their overall well-being, parenting skills, and caregiving responsibilities are recommended. In addition, implementing strategies to moderate the promotion of healthy behaviors and enhance opportunities for social support may also prove beneficial for parents.

## Figures and Tables

**Table 1 healthcare-12-00455-t001:** Participants’ demographic characteristics.

	Ν	%
**Gender**		
Males	23	18.6
Females	100	81.3
**Age ***	47.3 (6.3)	
**Marital status**		
Married (individuals)	75	60.9
Married (10) couples	20	16.2
Divorced	23	18.9
Unmarried	1	0.81
Widows	3	2.5
**Children** (either with ASD or not)		
1	33	26.8
2	72	58.5
>2	18	14.6
**Educational level**		
Secondary School	18	14.6
High School	52	42.2
University	32	26.0
MSc/Ph.D.	18	14.6
**Employment status**		
State employee	37	30.0
Private employee	49	39.8
Unemployed	21	17.0
Household	4	3.25
Retired	6	4.87
Freelance	5	4.2
**Faith toward God**		
Not at all close	4	3.25
Too little close	29	23.5
Little close	42	34.1
Close	12	9.75
Very close	36	29.2

* Mean (standard deviation).

**Table 2 healthcare-12-00455-t002:** The descriptive characteristics of the scales.

Scales	Mean	SD *	Min	Max
**FAS**				
Physical Fatigue (theoretical range 5–25)	20.81	5.80	8	32
Mental Fatigue (theoretical range 5–25)	8.92	1.60	5	13
**Total FAS** (theoretical range 10–50)	28.41	7.51	11	43
**MSPSS**				
Significant Others (theoretical range 1–7)	5.23	1.20	1	7
Family (theoretical range 1–7)	5.12	1.31	1	7
Friends (theoretical range 1–7)	5.01	1.29	1	7
**Total MSPSS** (theoretical range 1–7)	5.16	1.12	1	7
**FACIT Sp-12**				
Peace (theoretical range 0–16)	12.91	3.01	3	16
Meaning of Life (theoretical range 0–16)	8.80	4.01	2	16
Faith (theoretical range 0–16)	9.89	3.91	1	16
**Total FACIT Sp-12** (theoretical range 0–48)	31.51	8.73	10	46

* Standard Deviation.

**Table 3 healthcare-12-00455-t003:** The correlations between fatigue, social support, and spirituality.

Scales	Physical Fatigue	Mental Fatigue	Total FAS	Significant Others	Family	Friends	Total MSPSS
Significant Others	−0.40 (<0.001)	−0.21 (0.02)	−0.46 (<0.001)		0.86 (<0.001)	0.62 (<0.001)	0.94 (<0.001)
Family	−0.41 (<0.001)	−0.29 (<0.001)	−0.47 (<0.001)			0.54 (<0.001)	0.91 (<0.001)
Friends	−0.38 (<0.001)	−0.24 (0.007)	−0.40 (<0.001)				0.81 (<0.001)
Total MSPSS	−0.45 (<0.001)	−0.28 (<0.001)	−0.50 (<0.001)				
Meaning of Life	−0.34 (<0.001)	−0.14 (0.1)	−0.38 (<0.001)	0.54 (<0.001)	0.47 (<0.001)	0.31(<0.001)	0.50 (<0.001)
Peace	−0.55 (<0.001)	−0.31 (<0.001)	−0.56 (<0.001)	0.45 (<0.001)	0.56 (<0.001)	0.40 (<0.001)	0.53 (<0.001)
Faith	−0.21 (0.02)	−0.10 (0.2)	−0.24 (0.01)	0.22 (<0.001)	0.30 (0.01)	0.10 (0.5)	0.22 (0.01)
Total FACIT Sp-12	−0.46 (<0.001)	−0.24 (0.007)	−0.49 (<0.001)	0.49 (<0.001)	0.55 (<0.001)	0.32 (<0.001)	0.51 (<0.001)

Values are expressed as Pearson’s correlation coefficient (*p*-value).

**Table 4 healthcare-12-00455-t004:** Multivariable linear regression analysis with total fatigue as the dependent variable and social support, spirituality, and demographic characteristics as independent variables.

Independent Variables	Standardized Coefficient Beta	Unstandardized Coefficient Beta	95% CI *		*p*-Value
			Lower	Upper	
Mothers compared to fathers	0.32	4.81	2.11	7.60	0.001
Peace	−0.43	−0.80	−1.14	−0.51	<0.001
Significant Others	−0.20	−1.35	−2.37	−0.46	0.008

The above variables explained 42% of the variability of total FAS. * Confidence interval.

**Table 5 healthcare-12-00455-t005:** Multivariable linear regression with physical fatigue as the dependent variable and social support, spirituality, and demographic characteristics as independent variables.

Independent Variables	Standardized Coefficient Beta	Unstandardized Coefficient Beta	95% CI *		*p*-Value
			Lower	Upper	
Mothers compared to fathers	0.24	4.30	1.61	6.90	0.002
Peace	−0.50	−0.71	−0.93	−0.41	<0.001

The above variables explained 37% of the variability of physical fatigue. * Confidence interval.

**Table 6 healthcare-12-00455-t006:** Multivariable linear regression with mental fatigue as the dependent variable and social support, spirituality, and demographic characteristics as independent variables.

Independent Variables	Standardized Coefficient Beta	Unstandardized Coefficient Beta	95% CI *		*p*-Value
			Lower	Upper	
Mothers compared to fathers	0.21	0.92	0.25	1.64	0.01
Educational level	0.20	0.34	0.08	0.51	<0.008
Family Support	−0.37	−0.36	−0.51	−0.10	0.003

The above variables explained 21% of the variability of mental fatigue. * Confidence interval.

**Table 7 healthcare-12-00455-t007:** Multivariable linear regression with total FACIT Sp-12 as the dependent variable and social support, and demographic characteristics as independent variables.

Independent Variables	Standardized Coefficient Beta	Unstandardized Coefficient Beta	95% CI *		*p*-Value
			Lower	Upper	
Educational level	−0.22	−1.61	−2.60	−0.51	0.005
Close faith toward God	0.43	2.50	1.41	3.72	<0.001

The above variables explained 18% of the variability of total FACIT Sp-12. * Confidence interval.

## Data Availability

The data presented in this study are available on request from the corresponding author. The data are not publicly available due to participants’ confidentiality.

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
