# Peer review of "Fatigue among Greek Parents of Children with Autistic Spectrum Disorder: The Roles of Spirituality and Social Support"

_healthcare, 2024, doi:10.3390/healthcare12040455_

Round 1
Reviewer 1 Report
Comments and Suggestions for Authors
Overall this presents as an interesting study on a previously poorly investigated topic. The Introduction provides a decent foundation for this work, with exceptions noted below. Methods need further development, and Results are difficult to follow. Some points in the Discussion are unwarranted given the analyses presented.
Abstract
· Some description of analysis is needed in the abstract.
· The actual means from assessments could be removed.
Introduction
· Line 26, the term ‘differentiated’ is not the correct term to use in this context
· Line 46: please describe the difference between how mothers and fathers experience child behavior problems
· In the conclusion for this section, clearly identify the gap in the literature.
Materials and methods
· Line 88: “Functional ASD”, “Special Vocational Gymnasiums and Lyceums of Attica” are not widely utilized terms across the globe; please define.
· Line 99: are “Special Vocational High Schools of Attica” the same as Special Vocational Gymnasiums? Please be consistent with word choice.
· How many programs were approached to participate?
· Did each parent complete the surveys once, or multiple times? Please make this clear.
· Line 121: what is meant by “exact score”? Is this the total score across the categories of fatigue? If only the total score held meaning, it is not appropriate to then present ‘data’ in results on these subcategories.
· The choice of a p value of 0.2 for correlation coefficient supports only a weak correlation; the use of this to justify subsequent linear regression must be justified.
Results
· How many of the participants were married to each other?
· Since a point is made in the Introduction that fathers and mothers respond differently to child behavior difficulties, was a sub-analysis considered to separate these groups by role when calculating correlations?
· Are all children in Table 1 children with autism or just the total number in the family?
· Based on the discussion points, additional analyses are essential.
· The multivariate linear regression models generated all require greater explanation.
Discussion
· Lines 209-212: The suggestion re a link between levels of parental fatigue and severity of child differences reflects data that was not presented in this paper. There is no description of the children at all, and the rationale for moderate level of fatigue being related to extent of autistic features is unwarranted.
· Lines 212-217 seems to repeat results; this is unnecessary.
· Line 258-259: no data is presented to support the statement that mother experience more fatigue than fathers. Either present this data or remove these comments.
· Lines 265-267: The link between mothers’ fatigue and a sense of guilt, child dependence and lack of resilience is similarly unwarranted. No data to this effect has been presented, and authors do not cite existing literature.
Comments on the Quality of English Language
Word choice in several places is problematic. Some terms will not be universally understood.
Author Response
Reply to Reviewer’s 1 comments
Abstract
Comment 1
Some description of analysis is needed in the abstract.
Reply: we replied in lines 15-17.
Comment 2
The actual means from assessments could be removed.
Reply: we removed the phrase: “The mean of total FAS was 28.4 (SD±7.5), of total FACIT Sp-12 was 31.5 (SD±8.7), and of total MSPSS 5.1 (SD±1.1).”
Introduction
Comment 3
Line 26, the term ‘differentiated’ is not the correct term to use in this context.
Reply: we replaced the word ‘differentiated’ with the word “burdened”, line 26.
Comment 4
Line 46: please describe the difference between how mothers and fathers experience child behavior problems
Reply: we replied in lines 46-50.
Comment 5
In the conclusion for this section, clearly identify the gap in the literature.
Reply: we replied in line 90-92.
Materials and methods
Comment 6
Line 88: “Functional ASD”, “Special Vocational Gymnasiums and Lyceums of Attica” are not widely utilized terms across the globe; please define.
Reply: we replied in lines 101, 102, 111-112. If the reviewer can suggest an alternative definition which is widely utilized, we accept his/her definition.
Comment 7
Line 99: are “Special Vocational High Schools of Attica” the same as Special Vocational Gymnasiums? Please be consistent with word choice.
Reply: we replied in lines 103 and 112.
Comment 8
How many programs were approached to participate?
Reply: We kindly ask for reviewer to explain about the programs. Which programs does he/she mean? School programs? Parents do not follow a program.
Comment 9
Did each parent complete the surveys once, or multiple times? Please make this clear.
Reply: we replied in line 121.
Comment 10
Line 121: what is meant by “exact score”? Is this the total score across the categories of fatigue? If only the total score held meaning, it is not appropriate to then present ‘data’ in results on these subcategories.
Reply: we removed “exact score” line 136. Data in results on these subcategories are not presented.
Comment 11
The choice of a p value of 0.2 for correlation coefficient supports only a weak correlation; the use of this to justify subsequent linear regression must be justified.
Reply: we replied in lines 168-171.
Results
Comment 12
How many of the participants were married to each other?
Reply: we replied in line 179-180.
Comment 13
Since a point is made in the Introduction that fathers and mothers respond differently to child behavior difficulties, was a sub-analysis considered to separate these groups by role when calculating correlations?
Reply: we replied in lines 249-253.
Comment 14
Are all children in Table 1 children with autism or just the total number in the family?
Reply: We replied in table 1.
Comment 15
The multivariate linear regression models generated all require greater explanation.
Reply: we replied in lines 170-172, 231-233,238,239,242-244.
Discussion
Comment 16
Lines 209-212: The suggestion re a link between levels of parental fatigue and severity of child differences reflects data that was not presented in this paper. There is no description of the children at all, and the rationale for moderate level of fatigue being related to extent of autistic features is unwarranted.
Reply: we have already replied in lines 111, 115-117 and, also we added text in lines 335-337.
Comment 17
Lines 212-217 seems to repeat results; this is unnecessary.
Reply: we removed the repeated text.
Comment 18
Line 258-259: no data is presented to support the statement that mother experience more fatigue than fathers. Either present this data or remove these comments
Reply: Tables 4, 5 and 6 present the data that mothers experience more fatigue than fathers.
Comment 19
Lines 265-267: The link between mothers’ fatigue and a sense of guilt, child dependence and lack of resilience is similarly unwarranted. No data to this effect has been presented, and authors do not cite existing literature.
Reply: We provided a reference in line 319. The phrase was added in the context of the Discussion to be enriched.
Thank you
Dr. Victoria Alikari, Lecturer, Department of Nursing, University of West Attica.
Reviewer 2 Report
Comments and Suggestions for Authors
This study examined the cross-sectional associations of social support and spirituality with the fatigue of parents of children with autistic spectrum disorder. The results indicated that social support and spirituality are significant predictors of fatigue among parents of children with autistic spectrum disorder.
This study had several flaws that limited its values.
1. The authors should keep in mind that the study design is cross-sectional and the temporal relationships between fatigue and social support and spirituality could not be determined. It is possible that parents with severe fatigue may lack the energy to attend religious activities or even doubt the existence of God.
2. Although this study might be the first study examining the cross-sectional associations of social support and spirituality with the fatigue of parents of children with autistic spectrum disorder, I did not find the special meanings of this study that were different with other groups of children with illnesses. For example, will the results be different in the parents of children with other developmental problems such as epilepsy, intellectual disability, ADHD, etc?
3. This study did not measure the severity of symptoms of autistic spectrum disorder. Comorbidities have not been evaluated yet. They definitely have roles in parental fatigue.
4. This study did not evaluate the moderating effects of demographic characteristics on the associations of social support and spirituality with the fatigue.
5. English writing warrants a major revision. For example, title: “children in autistic spectrum disorder” should be “children with autistic spectrum disorder;” “the role of spirituality…” should be “the roles of spirituality…” abstract: “…parents with children with autistic spectrum disorder” should be “…parents with children with autistic spectrum disorder.” “Demographic data were, also, recorded. Data analysis was applied using IBM SPSS 21.0” should be deleted.
Comments on the Quality of English LanguageThis study examined the cross-sectional associations of social support and spirituality with the fatigue of parents of children with autistic spectrum disorder. The results indicated that social support and spirituality are significant predictors of fatigue among parents of children with autistic spectrum disorder.
This study had several flaws that limited its values.
1. The authors should keep in mind that the study design is cross-sectional and the temporal relationships between fatigue and social support and spirituality could not be determined. It is possible that parents with severe fatigue may lack the energy to attend religious activities or even doubt the existence of God.
2. Although this study might be the first study examining the cross-sectional associations of social support and spirituality with the fatigue of parents of children with autistic spectrum disorder, I did not find the special meanings of this study that were different with other groups of children with illnesses. For example, will the results be different in the parents of children with other developmental problems such as epilepsy, intellectual disability, ADHD, etc?
3. This study did not measure the severity of symptoms of autistic spectrum disorder. Comorbidities have not been evaluated yet. They definitely have roles in parental fatigue.
4. This study did not evaluate the moderating effects of demographic characteristics on the associations of social support and spirituality with the fatigue.
5. English writing warrants a major revision. For example, title: “children in autistic spectrum disorder” should be “children with autistic spectrum disorder;” “the role of spirituality…” should be “the roles of spirituality…” abstract: “…parents with children with autistic spectrum disorder” should be “…parents with children with autistic spectrum disorder.” “Demographic data were, also, recorded. Data analysis was applied using IBM SPSS 21.0” should be deleted.
Author Response
Reply to the Reviewer’s 2 comments
Comment 1
The authors should keep in mind that the study design is cross-sectional and the temporal relationships between fatigue and social support and spirituality could not be determined.
Reply: We provide some indicative cross-sectional studies (in lines 99-100) in different populations which we added in the references..
- Kazukauskiene, N., Bunevicius, A., Gecaite-Stonciene, J., & Burkauskas, J. (2021). Fatigue, Social Support, and Depression in Individuals With Coronary Artery Disease. Frontiers in psychology, 12, 732795. https://doi.org/10.3389/fpsyg.2021.732795.
- Baetz, M., & Bowen, R. (2008). Chronic pain and fatigue: Associations with religion and spirituality. Pain research & management, 13(5), 383–388. https://doi.org/10.1155/2008/263751
- Grossoehme, D. H., Friebert, S., Baker, J. N., Tweddle, M., Needle, J., Chrastek, J., Thompkins, J., Wang, J., Cheng, Y. I., & Lyon, M. E. (2020). Association of Religious and Spiritual Factors With Patient-Reported Outcomes of Anxiety, Depressive Symptoms, Fatigue, and Pain Interference Among Adolescents and Young Adults With Cancer. JAMA network open, 3(6), e206696. https://doi.org/10.1001/jamanetworkopen.2020.6696
- Szatkowska, K., & Sołtys, M. (2019). Perceived social support, spiritual well-being, and daily life fatigue in family caregivers of home mechanically ventilated individuals. Roczniki Psychologiczne, 21(1), 53-68.
Comment 2
It is possible that parents with severe fatigue may lack the energy to attend religious activities or even doubt the existence of God.
Reply: We replied in lines 68-74.
Comment 3
“Although this study might be the first study examining the cross-sectional associations of social support and spirituality with the fatigue of parents of children with autistic spectrum disorder, I did not find the special meanings of this study that were different with other groups of children with illnesses. For example, will the results be different in the parents of children with other developmental problems such as epilepsy, intellectual disability, ADHD, etc?”
Reply: We replied in lines 116-118.
Comment 4
This study did not measure the severity of symptoms of autistic spectrum disorder. Comorbidities have not been evaluated yet. They definitely have roles in parental fatigue.
Reply: We replied in lines 116-118.
Comment 5
This study did not evaluate the moderating effects of demographic characteristics on the associations of social support and spirituality with the fatigue.
Reply: In our study, the demographic data was independent variables, not moderators. Demographic data cannot and does not make sense to be moderators.
Comment 6
English writing warrants a major revision. For example, title: “children in autistic spectrum disorder” should be “children with autistic spectrum disorder;” “the role of spirituality…” should be “the roles of spirituality…” abstract: “…parents with children with autistic spectrum disorder” should be “…parents with children with autistic spectrum disorder.” “Demographic data were, also, recorded. Data analysis was applied using IBM SPSS 21.0” should be deleted.
Reply: We wrote in the title “parents of children with autistic spectrum disorder: the roles…”
Also, we deleted the phrase “Demographic data were, also, recorded. Data analysis was applied using IBM SPSS 21.0”).
Thank you.
Dr. Victoria Alikari, Department of Nursing, University of West Attica.
Round 2
Reviewer 2 Report
Comments and Suggestions for Authors
Although the authors made the revisions, the basic problems in study design can not be resolved.
Comments on the Quality of English Languagethis manuscript warrants major revisions in English writing.
Author Response
Dear reviewer,
thank you for your comment "Although the authors made the revisions, the basic problems in study design can not be resolved".
Reply: In this comment, we have given an evidence-based response from the first round with bibliography (lines 207-208).
Kind regards
Dr. Victoria Alikari.